

# The impact of bath gas composition on the calibration of photoacoustic spectrometers with ozone at discrete visible wavelengths spanning the Chappuis band

Michael I. Cotterell,[1,2] Andrew J. Orr-Ewing,[3] Kate Szpek[2] Jim M. Haywood[1,2] and Justin M. Langridge[2]

[1]College for Engineering, Mathematics and Physical Sciences, University of Exeter, Exeter, UK, EX4 4QF
[2]Observation Based Research, Met Office, Exeter, UK, EX1 3PB
[3]School of Chemistry, University of Bristol, Bristol, UK, BS8 1TS

*Correspondence to*: Michael I. Cotterell (m.cotterell@exeter.ac.uk)

**Abstract.** Photoacoustic spectroscopy is a sensitive *in situ* technique for measuring the absorption coefficient for gas and aerosol samples. Photoacoustic spectrometers (PAS) require accurate calibration by comparing the measured photoacoustic response with a known level of absorption for a calibrant. Ozone is a common calibrant of PAS instruments, yet recent work by Bluvshtein et al. (2017) has cast uncertainty on the validity of ozone as a calibrant at a wavelength of 405 nm. Moreover, Fischer and Smith. (2018) demonstrate that a low $O_2$ mass fraction in the bath gas can bias the measured PAS calibration coefficient to lower values for wavelengths in the range 532 – 780 nm. In this contribution, we present PAS sensitivity measurements at wavelengths of 405, 514 and 658 nm using ozone-based calibrations with variation in the relative concentrations of $O_2$ and $N_2$ bath gases. We find excellent agreement with the results of Fischer and Smith at the 658 nm wavelength. However, the PAS sensitivity decreases significantly as the bath gas composition tends to pure oxygen for wavelengths of 405 and 514 nm, which cannot be rationalised using arguments presented in previous studies. To address this, we develop a model to describe the variation in PAS sensitivity with both wavelength and bath gas composition that considers Chappuis band photodynamics and recognises that the photoexcitation of $O_3$ leads rapidly to the photodissociation products $O(^3P)$ and $O_2(X, v > 0)$. We show that the rates of two processes are required to model correctly the PAS sensitivity. The first process involves the formation of vibrationally excited $O_3(\tilde{X})$ through the reaction of the nascent $O(^3P)$ with bath gas $O_2$. The second process involves the quenching of vibrational energy from the nascent $O_2(X, v > 0)$ to translational modes of the bath gas. Both of these processes proceed at different rates in collisions with $N_2$ or $O_2$ bath gas species. Importantly, we show that the PAS sensitivity is optimised for our PAS instruments when the ozone-based calibration is performed in a bath gas with a similar composition to ambient air and conclude that our methods for measuring aerosol absorption using an ozone-calibrated PAS are accurate and without detectable bias. We emphasise that the dependence of PAS sensitivity on bath gas composition is wavelength dependent and we recommend strongly that researchers characterise the optimal bath gas composition for their particular instrument.



## 1 Introduction

The shortage of measurements of aerosol optical properties for light absorbing aerosol precludes their accurate representation in climate models (Stier et al. 2007). In particular, the light absorption for a particular class of aerosol referred to as *brown carbon* is poorly known, giving large uncertainties in the impact of brown carbon on climate (Feng et al. 2013; Lin et al. 2014).

Moreover, light absorption by brown carbon depends strongly on wavelength, with larger absorption at short (~400 nm) compared to longer (~700 nm) visible wavelengths. Therefore, the development of improved instruments for accurate aerosol absorption coefficient ($\alpha_{abs}$) measurements over the visible spectrum – particularly at short visible wavelengths relevant to brown carbon studies - is crucial to reduce the uncertainties in absorbing aerosol optical properties. Photoacoustic spectroscopy is a sensitive technique for measuring $\alpha_{abs}$ *in situ* for analytes that include gaseous or aerosol samples (Miklós et al. 2001,

Moosmüller et al. 2009). Our research focusses on developing PAS instruments to provide accurate and sensitive measurements of aerosol absorption coefficients in both laboratory and field studies (Davies et al. 2018; Lack et al. 2012).

To characterise aerosol in the natural environment, field-deployable instruments need to be both robust and compact. For example, we often operate our instrument aboard the UK research aircraft (FAAM BAe-146) which imposes constraints on

the instrument weight and dimensions. Traditional filter-based techniques for aerosol absorption measurement use a photo-detector to record the light transmission through a filter substrate on which ambient aerosol is impacted. These instruments are lightweight, robust and can operate over long periods (~days) unattended (Cappa et al. 2008; Sedlacek and Lee. (2007); Virkkula et al. 2005). However, there are known biases in the retrieved aerosol absorption coefficient; for example, Lack et al. (2008) report biases in the range 50 – 80%. These biases are attributed to several processes that include the modification of

the filter substrate by liquid aerosol components, changes in the aerosol structure and size upon impaction (e.g. from the redistribution of organic components and the aggregation of particles), and multiple scattering interactions. Filter-based absorption measurements are limited by their inability to study aerosol *in situ*.

PAS uses a laser beam to heat (by photoexcitation) the analyte aerosol *in situ* and the heated sample cools by collisional

relaxation with the bath gas. The bath gas consequently undergoes adiabatic thermal expansion and generates an acoustic pressure wave for detection by a sensitive microphone. The microphone response is directly proportional to $\alpha_{abs}$ and therefore provides an *in situ* measure of aerosol absorption. The relationship between this microphone response and the absorption coefficient is determined by calibrating the PAS with a calibrant of known or independently measured absorption, with the quality of this calibration determining the accuracy of PAS absorption measurements.


Researchers have used a variety of analytes to calibrate their PAS systems, including gas or aerosol calibrants (Bluvshtein et al. 2017; Davies et al. 2018; Fischer and Smith. 2018; Lack et al. 2006, 2012). For aerosol absorption measurement applications, an aerosol calibrant may be desired (Bluvshtein et al. 2017; Haisch. 2012). However, there can be large



uncertainties in aerosol-based calibrations that rely on a known refractive index for the aerosol, and accurate measurements of the size distribution and number concentration of the aerosol passed to the PAS. In particular, typical biases in number concentration are often quoted to be as large as 10% (Miles et al. 2011). Moreover, aerosol based calibrations require additional equipment such as a differential mobility analyser that are not deployed conveniently in the field. Therefore, many researchers

use a gas species to calibrate their PAS. Such calibrations typically pass the gaseous sample through the PAS to record the microphone signal $S$ before measurement of the gas extinction coefficient ($\alpha_{ext}$) by cavity ring-down spectroscopy (CRDS) using an in-line spectrometer. CRDS measures $\alpha_{ext}$ *directly* (without calibration) from the change in attenuation rate for transmitted light through a multi-pass optical cavity and, in the case of a gas species for which scattering can be assumed negligible, $\alpha_{ext}$ is equivalent to $\alpha_{abs}$. Therefore, the CRDS-measured $\alpha_{abs}$ is related to $S$ *via* the PAS calibration coefficient

(often referred to as the sensitivity) by:

$$C = \frac{S}{\alpha_{abs}}$$

Both $NO_2$ and ozone are popular analytes for calibration of PAS instruments (Bluvshtein et al. 2017; Davies et al. 2018; Fischer and Smith. 2018; Lack et al. 2006, 2012). The $O_3$ absorption cross section varies by only two orders of magnitude over the wavelength range of our spectrometers (405 – 658 nm), while the $NO_2$ cross section varies by three orders of magnitude. The

large $NO_2$ absorption cross section range causes saturation in the 405-nm spectrometers for concentrations that are optimal for the 658-nm spectrometers, preventing the fast (~1 hour) and simultaneous calibration of all our photoacoustic spectrometers from a single source of calibration gas. Therefore, we use ozone to calibrate our PAS instruments.

Lack et al. (2006) calibrated their 532-nm PAS with ozone and demonstrated a precision in the calibration coefficient of 0.09%.

Moreover, subsequent measurements of $\alpha_{abs}$ for nigrosin aerosol were in excellent agreement with expected values calculated from the known aerosol refractive index, controlled particle size and measured number concentration. However, Bluvshtein et al. (2017) repeated the experiments of Lack et al. (2006) using a PAS wavelength of 405 nm and found an unaccounted-for factor of two discrepancy between ozone-based and aerosol-based calibrations. This result challenged the validity of ozone as a calibrant for photoacoustic spectrometers at short wavelengths, i.e. the wavelength range that is often of most interest to

studies of brown carbon. We repeated the experiments of Bluvshtein *et al.* using our photoacoustic spectrometers that operate at wavelengths of 405, 514 and 658 nm, and instead observed excellent agreement between ozone and aerosol calibrations across all wavelengths, including the short 405 nm wavelength (Davies et al. 2018).

Most recently, Fischer and Smith. (2018) recognised that one key difference between the ozone calibrations performed by

ourselves and by Bluvshtein *et al.* was the composition of the bath gas. Bluvshtein et al. (2017) diluted an ozonised oxygen flow with $N_2$ to give an overall bath gas composition of 90% $N_2$ and 10% $O_2$, while we (Davies et al. 2018) injected an ozonised oxygen flow into ambient air to give an overall bath gas composition of 76% $N_2$ and 24% $O_2$ (ignoring <1% concentrations of



argon, $CO_2$ and trace gases). Fischer and Smith. (2018) measured the PAS calibration coefficient as a function of the $O_2$:$N_2$ ratio for PAS wavelengths of 532, 662 and 780 nm. They showed that the PAS calibration coefficient depends on $O_2$ mole fraction, with a factor of two increase in the PAS calibration coefficient as the bath gas $O_2$ mole fraction increased from 0.0 to 1.0, reaching an asymptotic maximum value that agreed with calibration coefficients measured using $NO_2$ gas or an aerosol

calibrant. We highlight three important aspects of the work of Fischer and Smith. First, the asymptotic value in the calibration coefficient is attained at an $O_2$ mole fraction of ~0.2 within measurement uncertainty, i.e. at the $O_2$ mole fraction that is relevant to our previous work (Davies et al. 2018). Second, the PAS sensitivity is reduced to only ~20% of the asymptotic value at the $O_2$ mole fraction of 0.1 that pertains to Bluvshtein et al. (2017). Therefore, the impact of bath gas does not reconcile our past measurements with those of Bluvshtein *et al.*, if the observed variations in PAS sensitivity with bath gas composition at

wavelengths >532 nm also apply at 405 nm. Third, the authors account for the asymptotic behaviour by using a model that treats the relaxation of photoexcited ozone and the relative rates at which this excited state is quenched by either $O_2$ or $N_2$ bath gas species. Their model ascribed the drop in PAS sensitivity (i.e. the calibration coefficient) at low $O_2$ mole fractions to poorer quenching of photoexcited $O_3$ by $N_2$. The authors concluded that ozone calibrations should be performed in a bath gas of pure $O_2$.

One drawback of the Fischer and Smith. (2018) study is that the PAS sensitivity measurements were not performed for the 405 nm wavelength for which Bluvshtein et al. (2017) reported significant discrepancies between ozone and aerosol calibrations. This omission prevents a direct assessment of the contribution of bath gas to biases in the ozone calibrations performed in the Bluvshtein et al. (2017) study. A second drawback is that, in developing a model to describe their measured

PAS data, they attribute the generation of a photoacoustic signal to the direct relaxation of $O_3^*$ (the superscript * implying that $O_3$ is in an electronically excited state). However, as we discuss further in this contribution, $O_3$ photoexcitation within the Chappuis band (spanning an approximate wavelength range of 400 – 700 nm) gives photodissociation to $O(^3P)$ and $O_2(X)$ within <1 picosecond irrespective of bath gas. The nascent photofragments subsequently undergo further collisional reactions and relaxation. As we argue here, these collisional processes must be taken into account when interpreting the PAS response.

Here, we present measurements of the dependence of PAS sensitivity on the bath gas $O_2$:$N_2$ ratio at three wavelengths of 405, 514 and 658 nm that span the Chappuis band. Importantly, our measurements include the 405 nm wavelength at which our previous work and that of Bluvshtein *et al.* were performed (Bluvshtein et al. 2017; Davies et al. 2018). At short visible wavelengths, we report very different variations in the PAS sensitivity with bath gas $O_2$ mass fraction than those reported by

Fischer and Smith. (2018), although our measurements agree at the longer 658 nm wavelength. To describe our results and reconcile them with those of Fischer and Smith, we present a complete description of Chappuis band photodynamics, accounting for the photodissociation of ozone, and develop a PAS sensitivity model treating the relative rates of collisional reactions and energy quenching of the nascent photofragments in the presence of both $O_2$ and $N_2$ bath gas species. In the following section, we describe briefly our instrument and our method for performing ozone-based calibrations for different



bath gas compositions. Sect. 3 presents measurements of the variation in PAS sensitivity with bath gas composition, develops a model to describe the measured variations in PAS sensitivity, and justifies the determined relaxation rates of the nascent photofragments from Chappuis band photolysis in the context of previously published studies.

## 2. Experimental Methods and Data Processing

We have described our CRDS and PAS instrument in detail in a previous publication and only a brief overview of our spectrometers is provided here, with the reader directed to Davies et al. (2018) for further details. The following sub-sections describe the methods used to generate ozone, control the bath gas composition, measure the sample extinction using CRDS and measure the sample absorption using PAS.

### 2.1. Ozone generation

Figure 1(a) shows how we generated an ozone-laden oxygen flow that was then split to provide ozone to three flow lines that included PAS instruments operating at wavelengths 405, 514 or 658 nm. A mass flow controller (MFC) passed 0.15 standard litres per minute (SLPM) of high (99.999%) purity $O_2$ from a gas cylinder supply through an ozone generator that used a cold corona discharge. We varied the frequency of this discharge to control the ozone concentration in the output $O_3$-$O_2$ flow. For our calibrations, we chose ten values of discharge frequency in the range 3 – 800 Hz, with higher frequencies providing larger

ozone concentrations. The frequency of the ozone generator was controlled directly by a LabVIEW software interface. A manifold then split the ozone-laden sample to five lines, although only three of these ozone lines were used for direct injection of ozone into the bath gas that then passed to 405, 514 and 658-nm spectroscopy channels (with *channels* referring to spectrometers operating at different wavelengths, see Figure 1(c)). The output from the remaining two ozone lines were sent to an exhaust. Each ozone line had an approximate flow rate of 0.03 SLPM, although these rates were not exactly equal for all

lines and we discuss this further in Sect. 2.5. The ozone flows were passed through 0.125 inch Teflon tubing that minimised contamination.

### 2.2. Control of the bath gas composition

A schematic for the control of the bath gas composition is shown in Figure 1(b). The flow rate at point '4' in Figure 1(b) is determined by the total flow rate regulated by the MFCs in Figure 1(c), the flow rate of the ozonised flow and the flow rate of

purge gas used to prevent deposition of contaminants on the CRDS mirrors (see Sect. 2.3). An aluminium mixing vessel with a 1.1 L volume was used to mix ambient air with a controlled flow of either $O_2$ or $N_2$ from a high purity (99.999%) gas cylinder. A MFC was used to set the mass flow of either $O_2$ or $N_2$ into the mixing volume, with any make up flow consisting of ambient air (~21% $O_2$, 78% $N_2$, 1% argon and trace amounts of other gases) with an in-line HEPA filter used to remove aerosol. The mass flow rate of ambient air into the mixing volume was monitored with a mass flow meter, with the magnitude determined

by the total flow rate at position '4' and the controlled mass flow of $O_2$ or $N_2$ into the mixing volume. By changing the mass





flow rate of the high purity gas species, we varied the oxygen mass fraction of the bath gas over the range 0.0 – 1.0. The mixed gas was then passed through a Nafion dryer that dried the gas to a relative humidity <4% before passing to a $NO_X/O_3$ scrubber to remove contributions to gas phase absorption from trace bath gas species. Finally, the bath gas passed through a further HEPA filter. We confirmed that the bath gas was devoid of particles by using a condensation particle counter.

Figure 1(c) shows that the bath gas was split into three flow lines corresponding to the 405, 514 and 658-nm spectroscopy channels. Each ozonised flow was mixed with bath gas 1 cm prior to a spectrometer sample inlet. The total sample flow rates through the three spectroscopy flow lines were controlled by MFCs set to 1.0 L min$^{-1}$. This flow rate corresponded to mass flow rates of 0.97 SLPM at the 950 hPa pressure measured for all the experiments performed in this work.

Figure 1(c) shows that we operated CRDS cells at wavelengths of 405 and 658 nm and PAS cells at wavelengths of 405, 514 and 658 nm. For each channel, the sample (bath gas with added ozonised flow and CRDS purge gas) passed through both a CRDS and PAS channel that operated at the same wavelength. We chose to pass the sample through the CRDS channel first because its 0.8 s residence time was lower than for the PAS cell (~12 s). The low residence time in the CRDS channels

minimized ozone loss to surfaces between successive spectrometer measurements. Moreover, we maintained short (< 15 cm) lengths of 0.25 inch conductive tubing between the CRDS and PAS. No corresponding CRDS channel operating at the same wavelength was available for the 514-nm PAS and the sample passed through only the 514-nm PAS only. We now describe the operation of our cavity ring-down and photoacoustic spectrometers.

## 2.3. Cavity ring-down spectrometers

We used two CRD spectrometers with identical configurations, albeit using different laser sources and cavity mirrors optimised for two different wavelengths. The output from a continuous wave diode laser was injected into a high finesse optical cavity that consisted of two highly reflective mirrors (reflectivities of >99.999%) separated by a distance of 40 cm. The laser diode current was modulated with a 50:50 duty cycle square wave signal to pulse the laser power between 0 and ~300 mW at a frequency of 2000 Hz. The spectral widths of the lasers (~100 GHz) were much larger than the free spectral range of the CRDS

optical cavity (~375 MHz). Therefore, the laser coupled passively into the optical cavity and overlapped numerous longitudinal modes. For each pulse injected into the optical cavity, a fraction of light leaking from the rear mirror was detected with a photomultiplier tube, the voltage from which was recorded by a 2.5 mega-samples per second (MS/s) data acquisition (DAQ) card. The time trace in this voltage is referred to as the ring-down trace. In the case that the linewidth of the sample extinction is larger than the linewidth of light circulating in the optical cavity, a criteria met for the cases of ozone in the Chappuis band

and for aerosol, Zalicki and Zare. (1995) demonstrated that the time dependence in the cavity output intensity obeys a single exponential decay, with the characteristic $1/e$ folding time for this decay referred to as the ring-down time ($\tau_{RD}$). Therefore, by fitting the ring-down trace to a single exponential, $\tau_{RD}$ was determined. We used the linear regression of the sum method described by Everest and Atkinson. (2008) for the fast and accurate retrieval of $\tau_{RD}$ in real time. The cavity mirrors were



mounted on kinematic mirror mounts (Newport) and the alignment of each mirror was optimised to maximise both $\tau_{RD}$ and the maximum intensity (voltage) in the ring-down trace, thereby maximising the sensitivity of each CRDS channel.

Sample inlet and outlet ports were located at opposite ends of the cavity and 3 cm away from the CRDS mirrors. To prevent
contaminants depositing on the highly reflective mirror surfaces, a 0.0125 SLPM purge gas flow was passed over the mirror surfaces. Zero air was used as the purge gas for calibrations corresponding to a bath gas $O_2$ mole fraction of 0.23 (i.e. ambient conditions) only and represents the purge gas used in measurements (including calibrations) under normal operation of the instrument in the field or laboratory. Otherwise, high purity $N_2$ or $O_2$ purge flows were used.

From knowledge of the ring-down time in the presence ($\tau_{RD}$) and absence ($\tau_{RD,0}$) of the sample, the extinction coefficient $\alpha_{ext}$ is calculated using:

$$\alpha_{ext} = \frac{R_L}{c}\left(\frac{1}{\tau_{RD}} - \frac{1}{\tau_{RD,0}}\right)$$   2

in which $c$ is the speed of light and $R_L$ is the ratio of the cavity mirror separation (40 cm) to the length over which the sample occupies the cavity. We take this latter length to be the distance between the sample inlet and outlet ports (34 cm) and $R_L$ is
taken as 1.1765 for both CRDS channels. We applied a small correction to $\alpha_{ext}$ to account for the small dilution of the sample by the purge gas flows. Typical values of $\tau_{RD,0}$ are $24 \pm 0.004$ and $34 \pm 0.04$ µs for the 405-nm and 658-nm CRDS channels, with the quoted uncertainties corresponding to one standard deviation over 60 seconds.

For the calibrations in this work, the ozone extinction at the 514-nm PAS wavelength ($\alpha_{ext,514}$) was calculated from the 658-
nm CRDS-measured extinction ($\alpha_{ext,658}$) from knowledge of the variation in ozone cross section with wavelength using:

$$\alpha_{ext,514} = \alpha_{ext,658} \cdot \frac{\sigma_{O_3,514}}{\sigma_{O_3,658}}$$   3

in which $\sigma_{O_3,514}$ and $\sigma_{O_3,658}$ are the $O_3$ absorption cross sections at 514 and 658 nm, respectively. We used the recommended absorption cross sections for $O_3$ provided by the NASA Jet Propulsion Laboratory (Burkholder et al. 2015). We also applied a further correction to $\alpha_{ext,514}$ to account for the difference in ozone concentrations between the 514-nm PAS and the 658-nm
CRDS due to the parallel flow configurations for these two spectrometers. This correction is described in Sect. 2.5. We can also calculate $\alpha_{ext,514}$ from the 405-nm CRDS measurements using the same approach above. However, Sect. 3.1 shows that there is significant uncertainty in the $\alpha_{ext,514}$ arising from uncertainty in the 405-nm laser wavelength. Therefore, this work only presents $\alpha_{ext,514}$ data calculated from 658-nm CRDS measurements.



## 2.4. Photoacoustic spectrometers

The output from a continuous wave diode laser was directed into an astigmatic multi-pass optical cavity that provided multiple reflections (~50) of the laser beam through a photoacoustic cell (PAS cell). The intensity of the laser beam was periodically modulated with a frequency that matched the resonance frequency of the PAS cell (see below). A photodiode behind the rear

cavity mirror monitored the RMS laser power, $W_{RMS}$. The geometry of the cell has been described previously by Lack et al. (2012) and consisted of two cylindrical resonators (an upper and lower resonator) that were coupled through acoustic buffer volumes. We used Brewster-angled windows to minimise the detection of laser interactions with the PAS cell windows and improve sensitivity. The laser beam was multi-passed through the lower resonator of the PAS cell. Sample inlet and outlet ports were located in opposite acoustic buffer volumes and the sample flow was drawn through the PAS cell. Ozone passing

through the laser beam was photoexcited and the bath gas was heated through collisional energy transfer from the ozone photoproducts to translational degrees of freedom of the bath gas. The heat in the bath gas generated a pressure (acoustic) wave through adiabatic expansion. These pressure waves coupled into a standing wave pressure eigenmode of the PAS cell, with the amplitude of the excited eigenmode detected by sensitive microphones located in each resonator. The voltage from each microphone was passed through a differential amplifier and the amplified output sent to a DAQ card that recorded the

microphone waveform with a time resolution of 8 MS/s over a 1 s interval.

A speaker was located close to the microphone in the lower resonator, and was driven by a voltage waveform that, in the frequency domain, was a top hat distribution over the frequency range 1250 – 1650 Hz. At multiple intervals during the calibration routine using ozone, the speaker was used to excite the standing wave eigenmode of the PAS cell. The 1 s

microphone time trace was recorded and processed through a Fast Fourier Transform that gave an acoustic spectrum with a Lorentzian distribution (see Fig. 2 of Lack et al. (2012)). By fitting this measured distribution to a Lorentzian function, the cell resonance frequency $f_{res}$ and quality factor $Q$ were measured. Importantly, this measured $f_{res}$ was used to set the modulation frequency of the laser to ensure this frequency matched the PAS cell resonance frequency at all times.

To measure the PAS response from ozone absorption during calibrations, 1 s waveforms were recorded and the amplitude of the frequency component corresponding to $f_{res}$ was measured. This amplitude is referred to as the raw photoacoustic signal $S_{raw}$. As described in previous publications (Davies et al. 2018; Lack et al. 2012), $S_{raw}$ requires correcting for $f_{res}$, $Q$ and $W_{RMS}$ and it can be shown from first principles (Miklós et al. 2001) that this correction should be performed according to:

$$S_{corr} = S_{raw} \cdot \frac{f_{res}}{W_{RMS}Q} \qquad\qquad 4$$

in which $S_{corr}$ is the corrected PAS signal. In our measurements, $W_{RMS}$ is measured from the voltage response of an uncalibrated photodiode detector. Therefore, the units of $W_{RMS}$ and $S_{corr}$ are arbitrary. An additional correction is required to





account for a background contribution to $S_{corr}$ from laser interactions with the PAS cell windows. We denote this background contribution $S_{corr}^{bg}$. Each ozone calibration performed for this work lasted ~1 hour during which the ozone concentration was increased sequentially. Before and after this ~1 hour period when the PAS cells were devoid of any absorbing sample, the mean $S_{corr}^{bg}$ was characterised over 60 s periods. These two mean background values were identical within measurement precision and a linear interpolation between these values was used to describe $S_{corr}^{bg}$ over the calibration period. The PAS signal of interest is then given by:

$$S_{final} = S_{corr} - S_{corr}^{bg}$$ 5

For the 405-nm and 658-nm PAS channels that were in a serial flow configuration with a CRDS channel (see Figure 1(c)), an additional but small correction was applied to $S_{final}$ to account for dilution by the CRDS purge flows.

## 2.5. Correction of $\alpha_{ext,514}$ for differences in ozone concentrations

Sect. 2.3. stated that a correction to the calculated $\alpha_{ext,514}$ was applied to account for unequal ozone concentrations between the 514-nm PAS and 658-nm CRDS channels. We used our measurements of $f_{res}$ for the 514-nm and 658-nm PAS channels to calculate the relative difference in $O_3$ between flow lines. We demonstrated in Davies et al. (2018) that the resonance frequency shift $\Delta f_{res}$ upon the introduction of the ozonised flow could be used to correct $\alpha_{ext,514}$ according to:

$$\alpha_{ext,514}^{corr} = \alpha_{ext,514} \cdot \frac{\Delta f_{res,514}}{\Delta f_{res,658}}$$ 6

## 2.6. Calibration procedure and the calculation of PAS sensitivity

We measured the PAS calibration coefficient for multiple values of bath gas $O_2$ mass fraction. Each calibration involved measuring $S_{final}$ and corrected extinction coefficients for ten ozone concentrations (ten values of coronal discharge frequency) and the calibration procedure was controlled by automated LabVIEW software. For a given coronal discharge frequency, we waited 120 seconds for the new ozone concentration to stabilise across all channels. The $f_{res}$ and $Q$ for the PAS channels were measured at the end of this wait period. Then, $S_{raw}$ and $\alpha_{ext}$ were measured at 1 Hz and averaged over a 60 second period before the coronal discharge frequency was increased to a higher level. The $S_{raw}$ and $\alpha_{ext}$ for each ozone concentration were corrected using the procedures described in Sect. 2.3. – 2.5., giving the values $S_{final}$ and $\alpha_{ext}^{corr}$. PAS sensitivity was calculated from a linear regression of the variation in the mean $S_{final}$ with mean $\alpha_{ext}^{corr}$ for all ten ozone concentrations constrained through the intercept $S_{final} = 0$, consistent with the definition of PAS sensitivity in equation 1.





## 3. Results and discussion

### 3.1. Measured variations in PAS sensitivity with bath gas composition

For each PAS channel, Figure 2 shows example calibration plots for the variation in PAS response ($S_{final}$) with the CRDS-measured extinction ($\alpha_{ext}^{corr}$) for bath gas compositions with oxygen mass fractions ($x_{o_2}$) of $x_{o_2} = 0.04$ (near-pure N$_2$), 0.23

(ambient air with added ozonised O$_2$ flow) and 1.0 (pure O$_2$). The plot for the 514-nm PAS channel does not show data for the $x_{o_2} = 1.0$ calibration. This is because the derived 514 nm extinction relies on correcting the 658-nm extinction, measured by CRDS in a parallel flow configuration, for differing ozone concentrations caused by unequal splitting of the ozonised flow by the gas manifold (Figure 1(a)). As described in Sect. 2, a correction factor for this uneven ozone splitting is determined by the ratio of PAS resonance frequency shifts in the 514-nm and 658-nm channels, $\Delta f_{res,514}/\Delta f_{res,658}$, upon the addition of the

ozonised flow. In the case of a bath gas of $x_{o_2} = 1.0$, the resonance frequency shift in all PAS cells upon the addition of the ozonised flow is zero, thus precluding the calculation of $\Delta f_{res,514}/\Delta f_{res,658}$ and the 514-nm extinction.

In Figure 2, each data point corresponds to the mean values for measurements of $S_{final}$ and $\alpha_{ext}^{corr}$ over a 60 s period at a given corona discharge lamp frequency, with one standard deviation error bars shown; these uncertainties are barely visible on Figure

2 due to their low value. Typical 1σ uncertainties in the extinction and absorption measurements are 0.74% and 0.17% for α$_{ext-405}$ and α$_{ext-658}$ respectively, and 2.43%, 0.36% and 0.32% for $S_{final,405}$, $S_{final,514}$ and $S_{final,658}$ respectively. The 1σ uncertainty in α$_{ext-514}$ is related to that in α$_{ext-658}$, although there is additional uncertainty in the 514-nm extinction measurement arising from the correction factors for $\Delta f_{res,514}/\Delta f_{res,658}$ and differences in ozone cross section at the 514-nm and 658-nm laser wavelengths. Typically, the standard error in $\Delta f_{res,514}/\Delta f_{res,658}$ is 1.67%. Meanwhile, the uncertainty in O$_3$ cross section ratio

$\sigma_{O_3-514}/\sigma_{O_3-658}$ for a 1 nm wavelength uncertainty in the 658-nm laser wavelength is 1.8%. Therefore, the uncertainty in the measured α$_{ext-514}$ is 2.5%. For comparison, the uncertainty in the ratio $\sigma_{O_3-514}/\sigma_{O_3-405}$ for a 1 nm uncertainty in the 405-nm laser wavelength is 7.5% and would correspond to a 7.7% uncertainty in the measured α$_{ext-514}$. Hence, due to the sensitivity of the O$_3$ cross section ratios to uncertainties in the laser wavelength, we opt to use the 658-nm laser in calibrating our 514-nm PAS channel.

Figure 2 demonstrates the excellent linearity in calibrations for all channels and bath gas compositions over the extinction range covered. Each calibration is fit to a straight line *via* a least squares fit routine, with the intercept constrained to pass through zero. The slope of this linear fit is equal to the PAS sensitivity $C$ (see equation 1). For all calibrations performed for this publication, the mean standard errors in $C$ arising from the aforementioned least-squares fit routine are 0.31%, 0.33% and

0.40% for the 405, 514 and 658-nm PAS channels respectively.



Figure 2 also demonstrates that there are significant variations in $C$ with bath gas composition at all PAS wavelengths for the limited range of calibrations at different $x_{o_2}$ values presented. Meanwhile, Figure 3(a) shows all our measurements of PAS sensitivity with variation in $x_{o_2}$ for sixteen separate calibrations at fourteen different bath gas compositions. Our 658-nm measurements demonstrate similar behaviour to that reported by Fischer and Smith (Fischer and Smith. (2018)), approaching

a plateau as $x_{o_2}$ tends to one. However, the PAS sensitivities for wavelengths of 405 and 514 nm demonstrate a very different dependence on $O_2$ mass fraction. In both of these channels, the PAS sensitivity is a maximum at $O_2$ mass fractions close to 0.2 (i.e. at mass fraction values similar to that of ambient air) and is lower at other $O_2$ mass fractions.

To explain the measured variation in PAS sensitivity with bath gas composition and excitation frequency, we consider the

potential energy surfaces for ground and photoexcited states of $O_3$. Grebenshchikov et al. (2007) provide an excellent review of $O_3$ photodissociation over various bands that include the Chappuis band (wavelengths in the range of ~400 – 700 nm). The authors calculate the potential energy surfaces for $O_3$, including potential energy cuts along the $O_2$-O dissociation coordinate, and describe concisely the Chappuis band photo-dynamics. The Chappuis band arises from excitation to two adiabatic $^1A''$ states. The lower state correlates asymptotically to the formation of $O(^3P) + O_2(X^3\Sigma_g^-)$ and is a repulsive surface, while the

upper state is bound with a dissociation energy (corresponding to the formation of $O(^1D) + O_2(a^1\Delta_g)$) that cannot be overcome from Chappuis band excitation at room temperature. To support this latter point, the experiments of Levene et al. (1987) find no evidence for the formation of $O_2(a^1\Delta_g)$ state following Chappuis band photoexcitation. A symmetry-allowed conical intersection connecting the two $^1A''$ states is located close to the Franck-Condon point and the equilibrium bond length of the upper $^1A''$ state. After photoexcitation in the Chappuis band, the electronically-excited $O_3^*$ population is distributed equally

between the two electronic states as the two adiabatic transition dipole moments are similar in the Franck-Condon region. While $O_3^*$ in the lower repulsive state dissociates within a few tens of femtoseconds to form $O(^3P) + O_2(X^3\Sigma_g^-)$, $O_3^*$ population in the upper $^1A''$ state may undergo two vibrations at most before crossing the $^1A''$ conical intersection and dissociating *via* the lower repulsive state to $O(^3P) + O_2(X^3\Sigma_g^-)$ (Flöthmann et al. 1997). Nevertheless, the lifetime of the population in this upper state is less than a few picoseconds, demonstrated by the diffuse structures in the absorption cross section for $O_3$ within the

Chappuis band. Importantly, the ~$10^{-12}$ s timescale for photodissociation is much faster than the modulation period of our PAS laser intensity ($\tau_{mod}$ is in the range $600 – 780$ µs, corresponding to modulation frequencies of $1280 – 1640$ Hz) and can be considered an instantaneous process. Ultimately, Chappuis band excitation leads to the prompt photodissociation of $O_3(^1A'')$ and to the formation of $O(^3P)$ and $O_2(X^3\Sigma_g^-)$, in which the nascent $O_2$ photofragment is vibrationally excited ($v > 0$) (Flöthmann et al. 1998):

$$O_3 + hv(\lambda = 400 – 700 \text{ nm}) \rightarrow O(^3P) + O_2(X^3\Sigma_g^-, v > 0) \qquad 7$$




For PAS measurements of absorption by $O_3$, we need to consider the subsequent fate of $O(^3P)$ and $O_2(X^3\Sigma_g^-, v > 0)$ and, importantly, the rate at which energy is quenched into *translational modes* of bath gas $M = N_2$ and $O_2$ for the generation of a PAS signal.

First, we note that the nascent $O_2(X^3\Sigma_g^-, v > 0)$, although formed in the ground electronic state, is vibrationally excited (Flöthmann et al. 1998). For our PAS measurements, we need to consider the rates of quenching of energy from $O_2(X^3\Sigma_g^-, v > 0)$ to translational modes of both bath gas species $M = N_2$, $O_2$ through the quenching reaction:

$$O_2(X^3\Sigma_g^-, v) + M(v') \rightarrow O_2(X^3\Sigma_g^-, v\text{ - }1) + M(v') \qquad\qquad 8$$

Second, the nascent $O(^3P)$ rapidly combines with bath gas $O_2$ (collision frequency on the order of $10^{12}$ s$^{-1}$) to form ground state
$O_3(\tilde{X})$ *via* reactions 9 and 11:

$$O(^3P) + O_2(X^3\Sigma_g^-) \xrightarrow{k_1} O_3^{\ddagger}(\tilde{X}) \qquad\qquad 9$$

$$O_3^{\ddagger}(\tilde{X}) \xrightarrow{k_2} O(^3P) + O_2(X^3\Sigma_g^-) \qquad\qquad 10$$

$$O_3^{\ddagger}(\tilde{X}) + M \xrightarrow{k_3} O_3(\tilde{X}, v > 0) \qquad\qquad 11$$

in which M ($N_2$ or $O_2$) is a third body that removes energy from the initial metastable $O_3^{\ddagger}(\tilde{X})$; because the nascent $O(^3P)$ has significant kinetic energy following photodissociation, the initial $O_3^{\ddagger}(\tilde{X})$ has energy above the energy threshold correlating to $O(^3P) + O_2(X^3\Sigma_g^-)$. We emphasise that the resulting $O_3$ is now in the ground electronic state but is vibrationally excited ($v >$
0). The collisional stabilisation of $O_3^{\ddagger}(\tilde{X})$ (equation 11) competes with the re-dissociation of $O_3^{\ddagger}(\tilde{X})$ to $O(^3P) + O_2(X^3\Sigma_g^-)$ (equation 10), with this latter process expected to be fast compared to the stabilisation pathway. The contribution to the PAS signal from $O_3(\tilde{X}, v > 0)$ will depend on (i) the rate at which stabilised $O_3(\tilde{X}, v > 0)$ (through collision with $M = N_2$, $O_2$) is formed, and (ii) the rate of energy quenching from vibrationally excited $O_3(\tilde{X}, v > 0)$ into translational modes of $M = N_2$, $O_2$. For this latter process, Ménard-Bourcin et al. (1991) and Zeninari et al. (2000) measured vibration-to-translation (V-T) energy
transfer rates for both $N_2$ and $O_2$ bath species, with similar rates of ~$7.6\times10^5$ s$^{-1}$ atm$^{-1}$ that correspond to a characteristic timescale for quenching of ~1.4 μs at the 950 hPa pressure measured during our experiments. This timescale is sufficiently fast compared to our PAS modulation period that it can be considered instantaneous. We note that Ménard-Bourcin et al. (1991) and Zeninari et al. (2000) studied the V-T relaxation of ozone in a vibrationally excited state containing a single quantum of energy corresponding to a symmetric stretch ($v_1 = 1103$ cm$^{-1}$), bend ($v_2 = 701$ cm$^{-1}$) or asymmetric stretch ($v_3 =$
1042 cm$^{-1}$), i.e. $O_3$ was in the ($v_1,v_2,v_3$) state of (1,0,0), (0,1,0) or (0,0,1) prior to V-T relaxation. However, Siebert et al. (2002)





show that O₃ exists in vibrational states as high as (7,0,0) for energies below the O₃ photodissociation threshold (approximately 1 eV = 8065 cm⁻¹). Therefore, the relaxation of ozone in our studies that is formed in vibrational states close to the O-O₂ bond dissociation threshold will occur on a timescale longer than the 1.4 µs estimated above. However, we do not anticipate that the timescale for the removal of ~7 quanta of energy will be sufficiently larger than the 1.4 µs timescale for single quantum

relaxation as to impact on the PAS signal, and we assume the relaxation of O₃(X, $v > 0$) occurs much faster than the PAS laser modulation period. Thus, while we can assume that the quenching rate of vibrationally excited O₃($\tilde{X}$, $v > 0$) is essentially instantaneous on our PAS timescale, the rate of formation of O₃($\tilde{X}$, $v > 0$) from processes 9 - 11 for bath gases M = N₂, O₂ are not well studied. We expect the process of O₃($\tilde{X}$, $v > 0$) formation to impact on the measured PAS sensitivity if the associated formation rate is slow. Indeed, this recombination process is likely to proceed at a slower rate than that for the quenching of

the stabilised O₃($\tilde{X}$, $v > 0$); the collisional stabilisation of O₃$^{\ddagger}$($\tilde{X}$) to form O₃($\tilde{X}$, $v > 0$) (equation 11) competes with re-dissociation of O₃$^{\ddagger}$($\tilde{X}$) to O(³P) + O₂ (equation 10). This latter process likely proceeds on a timescale of tens of picoseconds at most.

We now develop a model to describe our PAS sensitivity. We will show that processes with associated timescales that are slow

compared to the PAS laser modulation period degrade the PAS sensitivity. Therefore, while the reactions of (7) for the production of O(³P) + O₂(X³Σ$_g^-$, $v > 0$) and for the energy quenching of the stabilised O₃($\tilde{X}$, $v > 0$) can be considered instantaneous, the rates of V-T relaxation of O₂(X³Σ$_g^-$, $v > 0$) and the rate at which stabilised O₃($\tilde{X}$, $v > 0$) is formed could be slow.

### 3.2. A PAS sensitivity model that considers energy transfer rates from both O(³P) and O₂(X³Σ$_g^-$, $v > 0$)

Using the model of Kosterev et al. (2006), the photoacoustic signal $C$ is described by (Arnott et al. 2003; Moosmüller et al. 2009):

$$C = \frac{C_0}{\sqrt{1 + (2\pi f_{res}\tau)^2}} \qquad 12$$

in which $f_{res}$ is the frequency of the laser modulation that is set to the resonant frequency of the PAS cell, $\tau$ is the relaxation time of the excited state and $C_0$ is the PAS sensitivity in the limit $\tau \ll (2\pi f_{res})^{-1}$, i.e. quenching is fast compared to the

modulation period of the laser intensity. In the current case of PAS measurements of O₃, a contribution to the total PAS signal is made from quenching of energy from both O(³P) and O₂(X³Σ$_g^-$, $v > 0$), and we write:

$$C_{total} = \frac{C_O}{\sqrt{1 + \left(2\pi f_{res}\tau_O\right)^2}} + \frac{C_{O_2^*}}{\sqrt{1 + \left(2\pi f_{res}\tau_{O_2^*}\right)^2}} \qquad 13$$



Here, $O_2^*$ represents ground state $O_2$ in a vibrationally excited state. As described above, the rate of quenching of energy from $O(^3P)$ is limited by the recombination of $O(^3P)$ with $O_2$, with stabilization by a further bath gas species of either $O_2$ or $N_2$. From equations 9 - 11, we can write rate equations for the production of the intermediate $O_3^{\ddagger}(\tilde{X})$ and, under the steady-state approximation, we find that the rate of $O_3(X, v > 0)$ production is given by:

$$\frac{d[O_3(X, v > 0)]}{dt} = \frac{k_1 k_3 [O(^3P)][O_2][M]}{k_2 + k_3[M]} \qquad 14$$

in which the rate constants $k_1 - k_3$ are those for equations 9 - 11, respectively. As discussed above, we expect the rate of collisional stabilisation of $O_3^{\ddagger}(\tilde{X})$ (proceeding with a rate constant $k_3$) to be slow compared to that for the re-dissociation pathway (proceeding with a rate constant $k_2$). In the limit $k_2 >> k_3[M]$, equation 14 simplifies such that the production rate of $O_3(X, v > 0)$ depends on $[O][O_2][M]$ and the rate at which $O(^3P)$ is lost is described by:

$$-\frac{d[O(^3P)]}{dt} = k_{O-O_2-N_2}[O(^3P)][O_2][N_2] + k_{O-O_2-O_2}[O(^3P)][O_2]^2 \qquad 15$$

in which $k_{O-O_2-N_2}$ and $k_{O-O_2-O_2}$ are rate coefficients related to the formation rate for $O_3^{\ddagger}(\tilde{X})$ ($k_1$), the re-dissociation of $O_3^{\ddagger}(\tilde{X})$ ($k_2$) and the quenching rate of $O_3^{\ddagger}(\tilde{X})$ to $O_3(X, v > 0)$ by M ($k_3$), for M = $N_2$ or $O_2$ bath gas species respectively. Thus, the time dependence in the loss of $O(^3P)$, and the formation of $O_3(\tilde{X}, v > 0)$, follows an exponential form with a time constant:

$$\tau_O = \left(k_{O-O_2-N_2}[O_2][N_2] + k_{O-O_2-O_2}[O_2]^2\right)^{-1} \qquad 16$$

15  Similarly, we write the characteristic relaxation time of $O_2(X^3\Sigma_g^-, v > 0)$ by:

$$\tau_{O_2^*} = \left(k_{O_2^*-N_2}[N_2] + k_{O_2^*-O_2}[O_2]\right)^{-1} \qquad 17$$

in which $k_{O_2^*-N_2}$ and $k_{O_2^*-O_2}$ are quenching rate constants in bath gas $N_2$ or $O_2$, respectively. We fit the model of equation (13) to our measured PAS data, using expressions for $\tau_O$ and $\tau_{O_2^*}$ provided in equations (16) and (17), respectively. For this fitting, the values of $k_{O-O_2-N_2}$, $k_{O-O_2-O_2}$, $k_{O_2^*-N_2}$ and $k_{O_2^*-O_2}$ are constrained such that their values are invariant with wavelength.

20  Although nascent products may be produced in different vibrational states for different photolysis energies, we find the aforementioned constraint on rate constants is necessary to reduce the uncertainties in fit parameters and give meaningful insight into the wavelength dependence of the PAS sensitivity to $O_3$ absorption. Also, we have reported previously (Davies et al. 2018) the excellent agreement between measured aerosol absorption (using an ozone calibration in a bath gas of ambient air) and predicted values, with a maximum discrepancy of 9%. Therefore, we expect the relaxation time constants $\tau_O$ and $\tau_{O_2^*}$

25  to be near-instantaneous relative to the PAS modulation frequency at ambient-air bath gas composition and $C_{total} = C_O + C_{O_2^*}$. Thus, in fitting the above model, we constrained the sum $C_O + C_{O_2^*}$ such that the maximum allowed value is 20% larger



than the measured maximum in $C_{total}$. The resonant frequency $f_{res}$ in equation 13 varies with the bath gas $O_2$ mass fraction and is taken as the mean measured cell resonance frequency during a calibration for a given bath gas composition. Figure 4 shows these measured variations in $f_{res}$ with $O_2$ mass fraction for each PAS channel, with $f_{res}$ decreasing by ~7 % as $x_{O_2}$ increases from $x_{O_2} = 0$ to $x_{O_2} = 1$. In contrast, the $f_{res}$ values vary by <2 Hz (<0.14%) over a single calibration as the $O_3$ concentration is

increased over the calibration period.

The ten fit parameters that include $k_{O-O_2-N_2}$, $k_{O-O_2-O_2}$, $k_{O_2^*-N_2}$ and $k_{O_2^*-O_2}$, in addition to six coefficients corresponding to $C_O$ and $C_{O_2^*}$ for each of our three PAS wavelengths, are fit to the measured data by minimising the sum of least-squares between measured and modelled values of PAS sensitivity. Figure 3 shows these best-fit model descriptions for each wavelength, while

Table 1 summarises the best-fit parameters. We fitted our model to measured data in the mass fraction domain and, therefore, the rate coefficients have units of $s^{-1}$. The model developed here describes the measured data very well, with the PAS signal suppressed at high $O_2$ mass fractions associated with a slow rate of quenching of $O_2(X^3\Sigma_g^-, v > 0)$ energy into translational degrees of freedom of $O_2$ bath gas molecules. The model agreement with the 514-nm PAS measurements is worse than that for the 405-nm and 658-nm channels and considerations of the larger errors associated with the 514-nm PAS measurements

(as described in Sect. 3.1.) cannot fully reconcile these differences. Instead, this poorer agreement is likely to be a consequence of the requirement to constrain the four rate coefficients to be invariant with wavelength; as we discuss below, different vibrational states of $O_2(X, v > 0)$ are accessed as the photolysis wavelength in reduced below ~550 nm that will affect relaxation rate constants. The calculated ratios $k_{O-O_2-O_2}/k_{O-O_2-N_2}$ and $k_{O_2^*-O_2}/k_{O_2^*-N_2}$ are also given in Table 1. Over all PAS wavelengths, the best fit of our model suggests that bath $O_2$ is more effective at stabilising the formation of $O_3(\tilde{X}, v > 0)$ by a

factor of ~11 compared to bath $N_2$. However, there is significant uncertainty in the fit values of $k_{O-O_2-N_2}$ and $k_{O-O_2-O_2}$, with a factor of two increase in either rate constant having little impact on the modelled PAS sensitivity. Reducing the uncertainty in this fit parameter requires more measurements of the PAS sensitivity at low $O_2$ mass fractions (<0.2) and reductions in the uncertainty in the $O_2$ mass fraction measurement. Meanwhile, bath $N_2$ is a factor of 1.68 more effective at quenching vibrational energy from $O_2(X^3\Sigma_g^-, v > 0)$ into translation modes compared to bath $O_2$. We also examine the relative contributions of the

$O(^3P)$ and $O_2(X^3\Sigma_g^-, v > 0)$ channels to the total PAS signal, with $C_{O_2^*}/C_O$ given in Table 1, while Figure 3(b) shows the contributions of both species to the total modelled PAS sensitivity. Figure 3(b) shows that the signal contribution from quenching of energy from $O(^3P)$ decreases to zero as the bath gas composition tends towards that of pure $N_2$. In this limit, the absence of $O_2$ prevents the formation of $O_3(\tilde{X})$ and the characteristic relaxation time in equation 16 tends to infinity. The $C_{O_2^*}/C_O$ ratio ranges from $1.3 - 4.1$ as the photolysis energy increases (wavelength decreases from 658 to 405 nm). This

suggests that the fraction of energy that goes into vibrational modes of the nascent $O_2(X)$, compared to that partitioning to $O(^3P)$ kinetic energy, increases with decreasing wavelength.





### 3.3. Understanding the best-fit rate constants for $O_2(X^3\Sigma_g^-, v > 0)$ and $O_3^\ddagger(\tilde{X})$

We begin by considering the quenching of $O_2(X^3\Sigma_g^-, v > 0)$ before considering that of $O_3^\ddagger(\tilde{X})$. First, we need to understand the vibrational energy distribution for the nascent $O_2(X^3\Sigma_g^-, v > 0)$; the relaxation rate of $O_2$ will depend on its vibrational state. Flöthmann et al. (1998) calculated the vibrational energy distribution of $O_2(X^3\Sigma_g^-)$ following the Chappuis band

photodissociation. At a photolysis wavelength of ~ 620 nm, the authors predicted that the vibrational energy distribution was Boltzmann-like, with the population of $v = 0$ dominating the distribution and agreeing well with the experiments of Levene et al. (1987). However, a shoulder in this distribution develops as the wavelength decreases; at wavelengths in the range 450 – 500 nm, a significant population of $v = 4 - 8$ is predicted and higher $v$ is accessed with decreasing wavelength. As the wavelength decreases further, we would expect $v > 8$ to be populated.

We now consider the rates of energy quenching from $O_2(X^3\Sigma_g^-, v > 0)$ to bath gas molecules and how these depend on $v$. Indeed, this quenching can occur through V-T and vibration-to-vibration (V-V) energy transfer; while the bath gas translation energy generates acoustic pressure waves relevant to a photoacoustic measurement, V-V energy transfer could influence the vibrational state from which V-T energy transfer occurs. Coletti and Billing. (2002) reported V-T and V-V rate constants for

$O_2(X^3\Sigma_g^-, v > 0)$ to $O_2$ bath gas for $v$ in the range 1 – 29. Meanwhile, Billing. (1994) reported the V-T rate constants for $O_2(X^3\Sigma_g^-, v)$ to $N_2$ bath gas for $v$ in the range 13 – 25, and Park and Slanger. (1994) measured the associated V-V rates. These studies provide data for a temperature of 300 K and the rate constants are plotted in Figure 5. We note that the total rate constants (the sum of the V-T and V-V rate constants) for an $O_2$ bath gas were validated by experimental measurements of Park and Slanger. (1994) and Hickson et al. (1998). In a bath gas of pure $O_2$ and for an initial nascent photoproduct $O_2(X^3\Sigma_g^-,$

$v)$ with $v \sim 8$ at wavelengths <500 nm, the V-V rate constant (~$6\times10^{-14}$ cm$^3$ s$^{-1}$) is approximately three orders of magnitude higher than the V-T transfer rate (~$2\times10^{-17}$ cm$^3$ s$^{-1}$). Therefore, V-V energy transfer dominates and rapidly quenches $O_2(X^3\Sigma_g^-,$ $v = m)$ (with $m > 0$) to $m \cdot O_2(X^3\Sigma_g^-, v = 1)$ $via$ single quantum transfer steps on a characteristic timescale of ~$0.4 - 0.6$ $\mu$s. Upon quenching all nascent $O_2(X^3\Sigma_g^-, v)$ to the $v = 1$ level, V-T energy transfer then becomes the only route to removing vibrational energy. For $v = 1$, the V-T rate is ~$5\times10^{-19}$ cm$^3$ s$^{-1}$ and corresponds to a characteristic quenching timescale of ~ 80 milliseconds,

i.e. ~$100\times$ slower than PAS laser modulation period. Conversely, in a pure $N_2$ bath gas and for an initial nascent photoproduct $O_2(X^3\Sigma_g^-, v)$ with $v \sim 8$, the V-V rate is less than the V-T rate. For $O_2$-to-$N_2$ V-V energy transfer, the two quantum transition $O_2(X^3\Sigma_g^-, v) + N_2(X, v' = 0) \rightarrow O_2(X^3\Sigma_g^-, v - 2) + N_2(X, v' = 1)$ is resonant at $v = 19$ and gives rise to the maximum in the V-V rate shown in Figure 5 (Park and Slanger. (1994)), but this rate decreases rapidly as $v$ departs from $v = 19$. At $v \sim 8$, the V-V rate is ~$2\times10^{-17}$ cm$^3$ s$^{-1}$ and the V-T rate is ~$8\times10^{-17}$ cm$^3$ s$^{-1}$, with the latter rate estimated from an exponential fit to V-T

rates available for $v > 13$ extrapolated to lower $v$ (see Figure 5). Assuming V-T energy transfer dominates at $v \sim 8$, energy is quenched into $N_2$ bath gas translational modes on a characteristic timescale of ~500 $\mu$s that is less (albeit only marginally) than the PAS modulation period (600 – 780 $\mu$s). We emphasise that there are large uncertainties in the V-T rates (Coletti and Billing.



(2002) stated that the accuracies are worse than 25%) and in the exact vibrational states of the initial $O_2(X^3\Sigma_g^-, \nu)$ photoproducts. However, it is encouraging that we can reconcile our measured decrease in PAS sensitivity as bath $O_2$ mass fraction increases with calculated V-V and V-T rates. These rates predict a similar V-T quenching timescale to the PAS laser modulation period in pure $N_2$, but much extended timescales (by a factor of ~100) in pure $O_2$. Moreover, Billing. (1994) noted

that the calculated V-T rates for $O_2(X^3\Sigma_g^-, \nu)$ are about a factor of two larger for $N_2$ bath gas compared to $O_2$, that is in good agreement with our measurements that suggest $k_{O_2^*-N_2}/k_{O_2^*-O_2}= 1.68$.

We now focus briefly on the quenching rates of $O_3^\ddagger(\tilde{X})$ by bath gas $O_2$ and $N_2$, and the observation that the best-fit $k_{O-O_2-O_2}/k_{O-O_2-N_2}$ is ~11. To the best of our knowledge, the quenching of the $O_3^\ddagger(\tilde{X})$ is ill studied and there are no past

measurements with which to compare our data. In studies of vibrational quenching of hot $O_3$ (below the dissociation threshold), Ménard-Bourcin et al. (1991) and Zeninari et al. (2000) reported these rates in $O_2$ and $N_2$ bath gases to be identical within measurement uncertainties. As discussed above, our model is relatively insensitive to $k_{O-O_2-N_2}$. Indeed, with further measurements of PAS sensitivity at lower $O_2$ mass fraction and with reductions in the uncertainties in $O_2$ mass fraction determinations (e.g. controlling the flow of the ozonised oxygen fraction into each channel directly with a MFC), $k_{O-O_2-N_2}$

should be retrieved to a higher accuracy and $k_{O-O_2-O_2}/k_{O-O_2-N_2}$ might be found to be closer to unity.

### 3.4. Other possible routes to degrading the PAS sensitivity

We explored other possible mechanisms to account for the variation in PAS sensitivity with bath gas composition. In particular, Greenblatt et al. (1990) reported that the $O_2$-$O_2$ dimer has five sharp absorption bands between 446 – 630 nm. However, none of these absorption bands are predicted to contribute to extinction or absorption at our spectroscopic wavelengths (see Fig. 1

of Greenblatt et al. (1990)). Moreover, the formation of the $O_2$-$O_2$ dimer increases strongly with increasing $O_2$ concentration, while our CRDS-measured extinction does not demonstrate any increase as the bath gas $O_2$ concentration increases. This can be seen in Figure 2, in which the maximum values in the CRDS-measured extinction show no dependence on the bath gas $O_2$ mass fraction. However, other authors should consider the influence of $O_2$-$O_2$ dimer absorption on their PAS measurements in the visible range, particularly when the CRDS measurement of extinction are performed at a different wavelength to that of

the PAS measurement of absorption. In particular, we note that dimer absorption at wavelengths of 477.3, 532.2, 577.2 and 630.0 nm correspond to significant absorption coefficients of 34.5, 5.5, 60.2 and 39.4 Mm$^{-1}$ at atmospheric temperature and pressure for a gas of pure $O_2$. Indeed, Fischer and Smith. (2018) used a PAS channel operating at 532 nm that is close to a dimer absorption feature at 532.2 nm, while the authors' CRDS measurements were performed at $\lambda = 658$ nm at which there is no dimer absorption contribution. The recommendation of Fischer and Smith that calibrations are performed in pure oxygen

could be detrimental at some PAS wavelengths due to effects associated with $O_2$-$O_2$ dimer formation, and we have shown in



this work that calibrations in pure oxygen are detrimental at wavelengths below ~600 nm associated with inefficient V-T quenching of $O_2(X, v > 0)$ by bath $O_2$.

## 4. Summary

We have studied the impact of bath gas composition on the PAS calibration coefficient determined using an ozone calibrant.
We varied the ratio of $O_2:N_2$ concentrations of the bath gas and measured the PAS sensitivity for three photoacoustic spectrometers that operate at wavelengths of 405, 514 and 658 nm. Our measured variation in PAS sensitivity with $O_2$ mass fraction at 658 nm is in excellent agreement with the measurements presented by Fischer and Smith. (2018). However, at the shorter wavelengths of 405 and 514 nm (higher photolysis energies), we find that the PAS sensitivity decreases as the $O_2$ mass fraction is increased above values of ~0.3 (i.e. the approximate composition of ambient air). We have developed a model to
explain these measured variations that fully accounts for the photodynamics of ozone in the Chappuis band. We find that the reduced sensitivity in the limit of pure $N_2$ corresponds to the inefficient recombination of $O(^3P)$ with bath gas $O_2$, while the reduced sensitivity in the limit of pure $O_2$ corresponds to the inefficient quenching of energy from $O_2(X, v > 0)$ into translational degrees of freedom of bath $O_2$ on the timescale of the PAS laser modulation period.

Importantly, we have demonstrated that the PAS sensitivity is optimised (i.e. biases are minimised) when the PAS calibration is performed in a bath gas with a composition corresponding to that of ambient air. In combination with the results of our previous publication demonstrating excellent agreement between expected and PAS-measured (using ozone calibrations with ambient bath gas compositions) aerosol absorption coefficient for a laboratory aerosol standard (Davies et al. 2018), we conclude that our methods for measuring aerosol absorption using an ozone-calibrated PAS are accurate and without detectable
bias.

Another important aspect to our work is that our calibrations were performed at the PAS wavelength of 405 nm for which we have previously demonstrated excellent agreement between an aerosol-based and ozone-based calibration (Davies et al. 2018), while Bluvshtein et al. (2017) find that the ozone calibration differs from an aerosol calibration by a factor of two. As discussed
above, the calibrations in our work are performed by injecting an ozonised flow into ambient air, while those of Bluvshtein *et al*. are performed in a bath gas composed of 10% $O_2$ and 90% $N_2$. Our measurements predict only a 2% difference in the PAS sensitivity at 405 nm for these two bath gas compositions. Therefore, we support the conclusion of Fischer and Smith. (2018) that the low bath $O_2$ mass fraction does not explain the poor ozone calibration results described by Bluvshtein *et al*.

We emphasise that the dependence of PAS sensitivity on bath gas composition is wavelength dependent within the Chappuis band, particularly for wavelengths in the range 400 – 660 nm, and researchers should perform measurements of their PAS sensitivity to ascertain the optimal bath gas composition for their instrument. Furthermore, researchers must consider the





impact of $O_2$ dimer absorption on their measurements, particularly when the PAS and CRDS measurements of absorption are performed at different wavelengths.

Finally, we note that some PAS instruments (including our own) are designed to operate on aircraft platforms and measure aerosol absorption at high altitude where the ambient pressure is reduced to values as low as 400 hPa. One aspect not considered in the present work is the impact of pressure on quenching rate coefficients and the impact this has on PAS sensitivity. Therefore, future work will study the impact of pressure on the PAS sensitivity variation with bath gas composition.

**Data availability**

For data related to this paper, please contact Michael I. Cotterell (m.cotterell@exeter.ac.uk) or Justin M. Langridge (justin.langridge@metoffice.gov.uk).

**Competing interests**

The authors declare that they have no conflict of interest.

**Acknowledgements**

This work was funded by the Met Office. Michael I. Cotterell and Jim M. Haywood thank the Natural Environment Research Council for support through the CLARIFY-2017 grant (NE/L013797/1). Michael I. Cotterell also acknowledges support from the Royal Society of Chemistry/Analytical Chemistry Trust Fund through a Tom West Fellowship. Further support was provided by the Research Council on Norway via the projects AC/BC (240372) and NetBC (244141). We thank Prof. Michael N.R. Ashfold (University of Bristol) for useful discussions concerning the potential impacts of $O_2$-$O_2$ dimer formation on the measurements presented in this work.

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





**Tables**

| PAS wavelength / nm | 405 | 514 | 658 |
|---|---|---|---|
| $C_O$ | 11.3 | 8.4 | 2.9 |
| $C_{O_2^*}$ | 46.6 | 14.0 | 3.8 |
| $k_{O-O_2-N_2}$ / s$^{-1}$ | $7.8 \times 10^4$ | | |
| $k_{O-O_2-O_2}$ / s$^{-1}$ | $8.6 \times 10^5$ | | |
| $k_{O_2^*-N_2}$ / s$^{-1}$ | $1.3 \times 10^4$ | | |
| $k_{O_2^*-O_2}$ / s$^{-1}$ | $7.9 \times 10^3$ | | |
| | | | |
| $C_{O_2^*}/C_O$ | 4.1 | 1.7 | 1.3 |
| $k_{O-O_2-O_2}/k_{O-O_2-N_2}$ | 11.1 | | |
| $k_{O_2^*-O_2}/k_{O_2^*-N_2}$ | 0.6 | | |

**Table 1: Summary of the best fit parameters for the PAS sensitivity model described in the main text.**

**Figures**

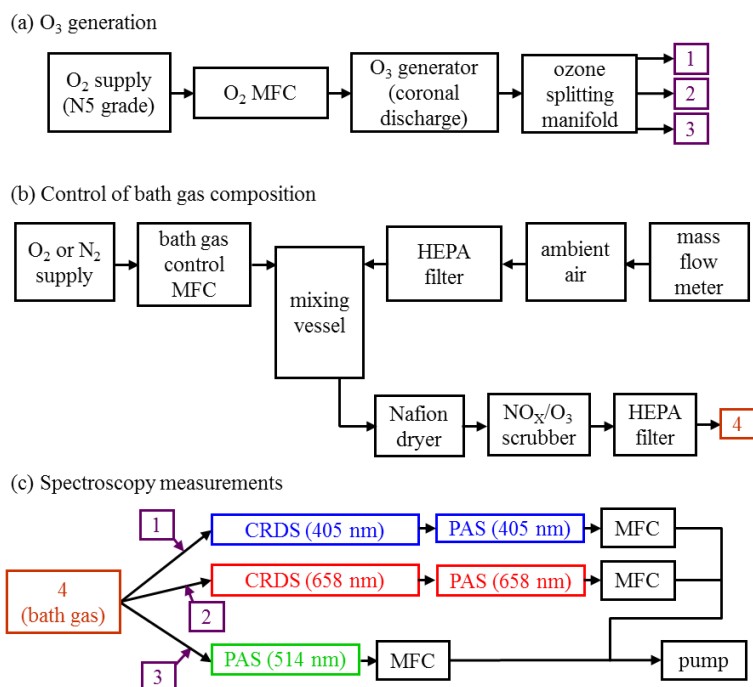

**Figure 1: Schematic diagrams of the experimental arrangements for (a) the generation of ozone, (b) controlling the composition of the bath gas, and (c) performing CRDS and PAS measurements of extinction and absorption, respectively, at different optical wavelengths. MFC denotes a mass flow controller.**



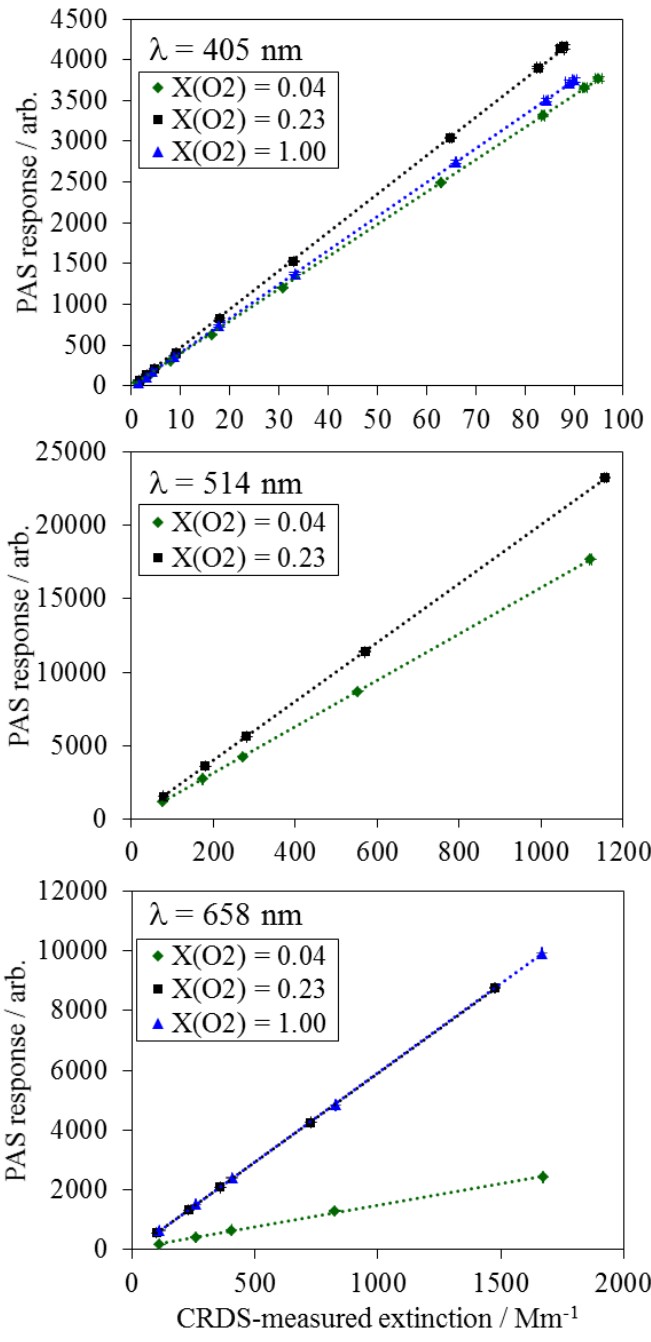

**Figure 2: For each PAS channel, example calibration measurements of PAS response variation with CRDS-measured extinction for an analyte of ozone gas. Each plot shows typical calibration data for ozone gas in bath gases composed of a mixture of $N_2$ and $O_2$, with $O_2$ mole fractions of 0.04 (near-pure $N_2$), 0.23 (ambient air composition) and 1.0 (pure $O_2$). All data points include error bars corresponding to one standard deviation in the measured PAS response or extinction, although these error bars are not visible on the plot scales because of the low variance in the measurements. Dashed lines represent straight line fits to the measured data, with the fit constrained such that the intercept is zero.**



**Figure 3: (a) The measured PAS sensitivity $C$ with variation in the bath gas $O_2$ mass fraction (points), and best fit descriptions of the data for the model described by equations 13, 16 and 17. (b) The best fit model description from (a) and the contributions from the two components of the model that correspond to quenching of $O(^3P)$ and $O_2(X^3\Sigma_g^-, v > 0)$.**





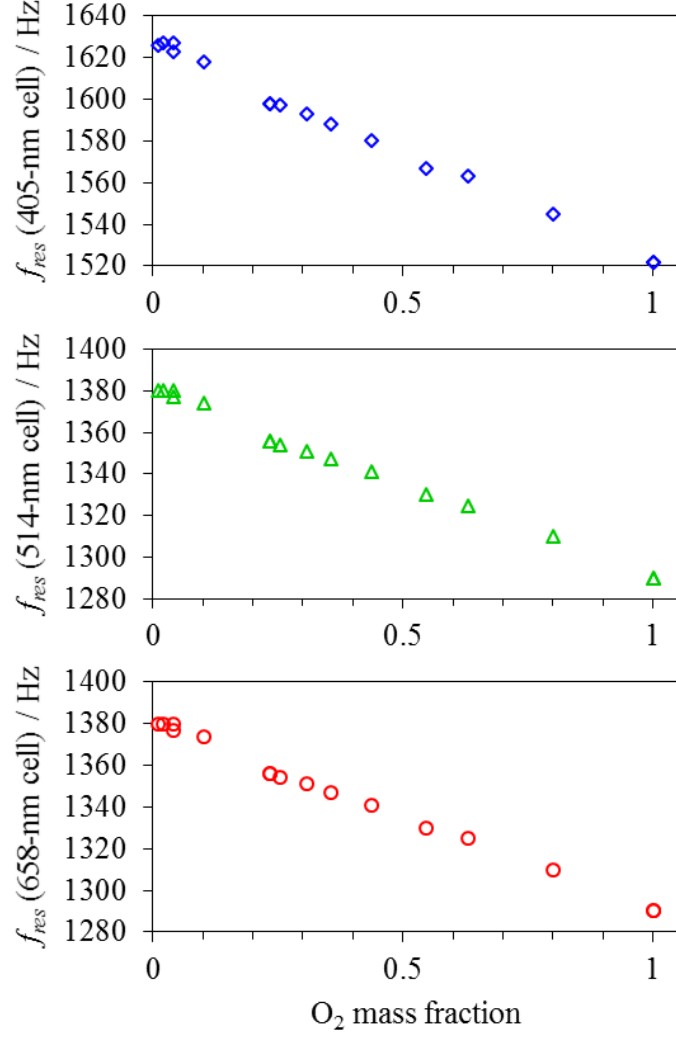

**Figure 4: The variation in the mean measured PAS cell resonance frequency with O₂ mass fraction for each PAS channel used in this study.**





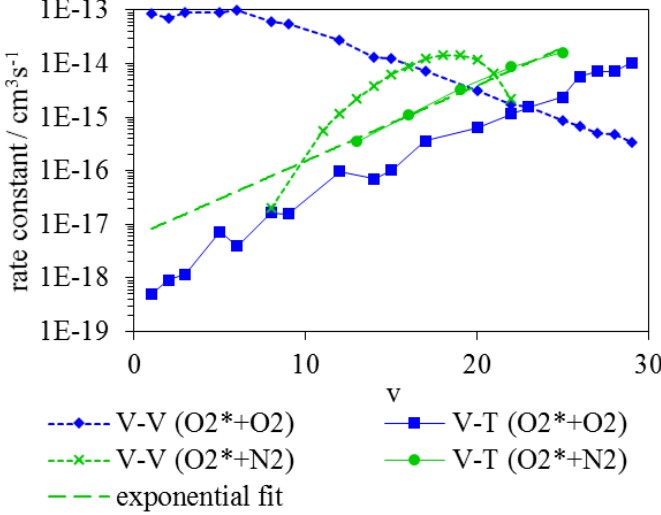

**Figure 5: Calculated quenching rates of $O_2(X^3\Sigma_g^-, v > 0)$ for V-T energy transfer to bath gas $O_2$ (blue squares) or $N_2$ (green circles), with variation in the vibrational quantum state. Also shown are the corresponding variations in V-V quenching rates of $O_2(X^3\Sigma_g^-, v > 0)$ in a bath gas of $O_2$ (blue diamonds) and $N_2$ (green crosses). Data are taken from references Billing. (1994); Coletti and Billing. (2002); Park and Slanger. (1994).**