# Peer review of "The impact of bath gas composition on the calibration of photoacoustic spectrometers with ozone at discrete visible wavelengths spanning the Chappuis band"

_Atmospheric Measurement Techniques, 2018_

## Referee Comment (RC1) · Anonymous Referee #1 · 28 Jan 2019

This manuscript focuses on the improvement of the calibration methodology of photoacoustic spectrometers (PAS). The paper settled some issues demonstrated by some previous studies when O3 is used as a calibrant, e.g., the uncertainty on the validity at a wavelength of 405 nm, and developed a model to describe the variation in PAS sensitivity with both wavelength and bath gas composition. The data presented in the manuscript is in general of good quality. The conclusions are sound and very useful for the researchers to better calibrate their PAS. The manuscript is very well written and organized and certainly meets the criteria of AMT. There is no more suggestions that I

can provide.
* * *

---

## Referee Comment (RC2) · Anonymous Referee #2 · 2 Mar 2019

This manuscript builds on previous work by Fischer and Smith (2018) and further explores the dependence of PAS sensitivity with ozone as calibrant on the ratio of N2 and O2 in the bath gas at relevant wavelengths, allowing for direct comparison with measurements by Bluvshtein et al. (2017). The paper is very well written. The experiment is well planned and carried out. The language is precise, and the analysis is thorough. I only have a couple of minor comments, as detailed below, for the authors to consider.

The authors calculated extinction coefficient at 514 nm from the extinction measurement at 658 nm since no direct measurements were available, but how reliable is the

calculation the what the associated uncertainty is? Have the authors tried calculating extinction coefficient at 405 nm from that at 658 nm and compare the results with the actual measurement of bext at 405nm? That way the accuracy of the calculated bext at 514 nm can be assessed.

Figure 3. Can the authors add vertical uncertainty bars for PAS sensitivity? \

Page 6 Line 17: . . . and the sample passed through only the 514-nm PAS only.

---

## Referee Comment (RC3) · Anonymous Referee #3 · 18 Mar 2019

Manuscript: The Impact of bath gas composition on the calibration of photo acoustic spectrometers with ozone at discrete visible wavelengths spanning the Chappuis band (Cotterell et al.,)

Referee Review of manuscript

High accuracy/precision measurements of aerosol light absorption continues to be a need as this measurement remains the controlling factor as realizing further reduction in the uncertainty in the aerosol direct radiative forcing contribution to global climate

change. To this end, in situ techniques have been developed to address measurement bias in filter-based measurements that are reaching their utility limits. Chief among these in situ techniques is the photo-acoustic spectrometer. By directly measuring light absorption through the photo thermal effect, the PAS can provide a high accuracy measurement of aerosol light absorption. The "rub" is having a high accuracy calibration. To this end, PAS calibrations have utilized diluted NO2 in air (or N2) and ozone in the RGB spectral range. However, with improvement in PAS measurement precision and because of growing need to better quantify the contribution of brown carbon (BrC) to aerosol absorption, new demands on improved measurement accuracy has arisen. But to address this need, improvement of the measurement accuracy of the calibration standard is needed. For example, while NO2 works quite well at the green wavelengths, a photodissociation pathway limits the utility of this cal gas at 405 nm (a popular emission wavelength). The present study aims at tackling this gap by furthering the work of Fischer and Smith (2018) using ozone at the RGB wavelengths by carefully defining the PAS calibration conditions on bath gas.

As one trained in chemical physics I particularly enjoyed the favor of this paper. As a matter of fact, this manuscript would easily fit in a journal of chemical physics. But, as the purpose of this work is to alert our aerosol community to the subtle but important consequences that bath gases have on energy transfer efficiency, and, in turn, on the overall accuracy of the PAS measurement, this manuscript is perfectly appropriate for AMT. This manuscript is very well written and thorough in its analysis, and thus deserves to be published. My comments are of a minor variety which can be readily corrected.

Page 2. Line 19: While discussing the measurement bias associated with the PSAP, the authors cite the Lack et al., study which reported a bias that ranges from ~50 - 80%. The authors could go a bit further and state that this bias exhibits an OA/BC dependence. While certainly not the focus of this paper, from a completeness point-of-view, citing this helps point to where some of the filter-based measurement bias likely

originates from.

Page 3; Line 15: The authors write "...large NO2 absorption cross section range causes saturation in the 405-nm spectrometers for concentrations..." This is a bit misleading. The primary issue for the use of NO2 at 405 nm is that a photodissociation pathway opens up at wavelengths $< \sim$ 420 nm. (See for example, Gardner et al., JGR 92 1987.) Please reword this to better reflect that photodissociation is what limits the use of NO2 as a cal standard at 405 nm. Besides, the reason put forth by the authors that one must use the same concentration to calibrate both the red and blue channels sounds more like a strawman argument.

Page 9, line 7: What is the nominal signal level of the background contribution (Sbgcorr) to the overall PAS signal. I'd like to have a sense of how great this background correction is. Presumably, since this contribution is constant irrespective of the aerosol loading, depending upon the background signal amplitude, this contribution could be more important at lower signal levels. Are we talking about Sbgcorr = 0.1 Mmˆ-1 or 1 Mmˆ-1 or 10s Mmˆ-1. I would like to have my fears allayed that in the limit of weak aerosol absorption signal (the atmospherically relevant situation) that the reported PAS signal for the aerosol is not the result of subtracting two big numbers to get a small number.

At the risk of being a bit pedantic, could you please reorder the legend on Figure 3b. As currently displayed, the traces are solid, dotted, and dashed, while the legend is solid, dashed, and dotted. This is certainly a cosmetic request, but makes it easily for the reader. Also, are error bars present on the PAS signal (and are just too small to be observed on the plot scale) or are they not present. If the latter, please add.

This reviewer is most intrigued by the potential consequences of reduced pressure at aircraft sampling altitudes on quenching rates. The authors kick this can down the road but this could be quite interesting from both a fundamental energy transfer perspective as well as a practical aspect. Staying with this theme, I cannot help but wonder if NO2

as a calibration gas - available with either N2 or air as the bath gas - might also exhibit a similar quenching sensitivities to differing bath gas mixtures as displayed by ozone. As highlighted above, one would expect that bath gas quenching could shift the quantum yield for NO2 photodissociation at 405 nm.

The last request, which the authors are encouraged to do, but certainly not required given the target audience. It would be nice to add a figure displaying the potential energy surfaces (even as a simple 2-D plot) for the various O3 dissociation pathways. This is certainly the chemical physicist in me requesting this as that is how I can readily see what is going on. For those researchers not accustomed to chemical physics, such a figure would help clarify the subtle pathways present in ozone.

---

## Author Response (AR1)

We respond to the comments made by the referees below with our responses highlighted in red.

Referee 1

This manuscript focuses on the improvement of the calibration methodology of photoacoustic spectrometers (PAS). The paper settled some issues demonstrated by some previous studies when O3 is used as a calibrant, e.g., the uncertainty on the validity at a wavelength of 405 nm, and developed a model to describe the variation in PAS sensitivity with both wavelength and bath gas composition. The data presented in the manuscript is in general of good quality. The conclusions are sound and very useful for the researchers to better calibrate their PAS. The manuscript is very well written and organized and certainly meets the criteria of AMT. There is no more suggestions that I can provide.

Response:
We thank the reviewer for reading our manuscript and for their supportive comments.

Referee 2

This manuscript builds on previous work by Fischer and Smith (2018) and further explores the dependence of PAS sensitivity with ozone as calibrant on the ratio of N2 and O2 in the bath gas at relevant wavelengths, allowing for direct comparison with measurements by Bluvshtein et al. (2017). The paper is very well written. The experiment is well planned and carried out. The language is precise, and the analysis is thorough. I only have a couple of minor comments, as detailed below, for the authors to consider.

Response:
We thank the reviewer for their very supportive comments. We have addressed the reviewer's minor comments below.

The authors calculated extinction coefficient at 514 nm from the extinction measurement at 658 nm since no direct measurements were available, but how reliable is the calculation the what the associated uncertainty is? Have the authors tried calculating extinction coefficient at 405 nm from that at 658 nm and compare the results with the actual measurement of $b_{ext}$ at 405nm? That way the accuracy of the calculated $b_{ext}$ at 514 nm can be assessed.

Response:
We have indeed calculated the 514-nm extinction using the measured $b_{ext}$ at 405 nm in addition to calculating 514-nm extinction from the $b_{ext}$ measurements at 658 nm. However, we discuss in the manuscript (see Sect. 3.1) that there is significant error in the calculated 514-nm extinction when we use the 405-nm $b_{ext}$ measurement caused by the 1 nm uncertainty in the 405 nm laser wavelength. This originates from the strong sensitivity in the ozone cross section to wavelength at short visible wavelengths, while this sensitivity is much reduced at long visible wavelengths approaching ~600 nm. The result of this strong sensitivity of ozone cross section to wavelength uncertainty is that the 514-nm extinction has associated uncertainties of 2.5% and 7.7% when calculated from the measured 658-nm and 405-nm $b_{ext}$, respectively. However, the reviewer is right that we did not compare in our analysis the 514-nm extinctions directly for calculations using either 405-nm or 658-nm measurements. Therefore, we have added on page 10, line 24 of the manuscript the following:

*For completeness, we find that the calculated $\alpha_{ext-514}$ from extinction measurements at the 405 and 658 nm wavelengths are well correlated (with a linear Pearson correlation coefficient of 0.93) and the average $\alpha_{ext-514}$ is 14% larger when calculated from 405-nm compared to 658-nm measurements.*

Figure 3. Can the authors add vertical uncertainty bars for PAS sensitivity?

Response:
These vertical error bars are already present on the PAS sensitivity plots (Figure 3). However, we had overlooked a statement to describe the error bars included in the figure legend. Therefore, in the figure legend for Figure 3, we have included the following statement:

*The measured data include vertical error bars that represent one standard deviation in the measured sensitivity, although these error bars are not visible on the vertical scale shown. Horizontal error bars represent the uncertainty in O₂ mass fraction arising from the standard errors in the mass flow controller flow rates that control concentrations of O₂ and N₂ in the bath gas.*

Page 6 Line 17: . . . and the sample passed through only the 514-nm PAS only.

Response:

We thank the reviewer for spotting this mistake. We have re-worded to say '*and the sample passed through the 514-nm PAS only'*.

Referee 3

High accuracy/precision measurements of aerosol light absorption continues to be a need as this measurement remains the controlling factor as realizing further reduction in the uncertainty in the aerosol direct radiative forcing contribution to global climate change. To this end, in situ techniques have been developed to address measurement bias in filter-based measurements that are reaching their utility limits. Chief among these in situ techniques is the photo-acoustic spectrometer. By directly measuring light absorption through the photo thermal effect, the PAS can provide a high accuracy measurement of aerosol light absorption. The "rub" is having a high accuracy calibration. To this end, PAS calibrations have utilized diluted NO2 in air (or N2) and ozone in the RGB spectral range. However, with improvement in PAS measurement precision and because of growing need to better quantify the contribution of brown carbon (BrC) to aerosol absorption, new demands on improved measurement accuracy has arisen. But to address this need, improvement of the measurement accuracy of the calibration standard is needed. For example, while NO2 works quite well at the green wavelengths, a photodissociation pathway limits the utility of this cal gas at 405 nm (a popular emission wavelength). The present study aims at tackling this gap by furthering the work of Fischer and Smith (2018) using ozone at the RGB wavelengths by carefully defining the PAS calibration conditions on bath gas. As one trained in chemical physics I particularly enjoyed the favor of this paper. As a matter of fact, this manuscript would easily fit in a journal of chemical physics. But, as the purpose of this work is to alert our aerosol community to the subtle but important consequences that bath gases have on energy transfer efficiency, and, in turn, on the overall accuracy of the PAS measurement, this manuscript is perfectly appropriate for AMT. This manuscript is very well written and thorough in its analysis, and thus deserves to be published. My comments are of a minor variety which can be readily corrected.

Response:

We are glad that the reviewer enjoyed reading our manuscript and we thank the reviewer for their very supportive comments. We address the reviewer's minor comments below.

Page 2. Line 19: While discussing the measurement bias associated with the PSAP, the authors cite the Lack et al., study which reported a bias that ranges from ∼50 - 80%. The authors could go a bit further and state that this bias exhibits an OA/BC dependence. While certainly not the focus of this paper, from a completeness point-of-view, citing this helps pint to where some of the filter-based measurement bias likely originates from.

Response:

We agree with the reviewer on this point. We have modified page 2 line 19 onwards to read:

*Lack et al. (2008) report biases in the range 50 – 80% with larger biases associated with aerosol samples containing a large organic fraction relative to black carbon, although we have demonstrated recently that advanced correction schemes can remove the bias dependence on organic mass fraction with modest biases in derived absorption coefficients of up to 17% (Davies et al. (2019)). Biases are also attributed to processes…*

Page 3; Line 15: The authors write ". . .large NO2 absorption cross section range causes saturation in the 405-nm spectrometers for concentrations. . ." This is a bit misleading. The primary issue for the use of NO2 at 405 nm is that a photodissociation pathway opens up at wavelengths < ∼ 420 nm. (See for example, Gardner et al., JGR

1987.) Please reword this to better reflect that photodissociation is what limits the use of NO2 as a cal standard at 405 nm. Besides, the reason put forth by the authors that one must use the same concentration to calibrate both the red and blue channels sounds more like a strawman argument.

Response:

We agree with the reviewer. In our initial manuscript, we decided to avoid discussing the $NO_2$ photolyis pathway to avoid giving the impression that photolysis of a gas strongly degrades its utility as a calibrant; indeed, $O_3$ photodissociates at the wavelengths used in our work. The key point is that $NO_2$ is lost irreversibly through photolysis (forming stable NO and $O_2$ products) and the nascent NO products are not efficiently quenched by air. On reflection, we have sided with the reviewer that we should state that the $NO_2$ photolysis pathway limits the use of $NO_2$ as a calibration gas and we have added the following statement after 'preventing the fast (~1 hour) and simultaneous calibration of all our photoacoustic spectrometers from a single source of calibration gas':

*Importantly, $NO_2$ photodissociates at optical wavelengths <430 nm, with $NO_2$ lost irreversibly from the sample to form stable nascent NO and $O_2$ products (Tian et al. 2013). This photodissociation pathway limits $NO_2$ as a*

*calibration standard for photoacoustic spectrometers at short visible wavelengths.*

Page 9, line 7: What is the nominal signal level of the background contribution (Sbgcorr) to the overall PAS signal. I'd like to have a sense of how great this background correction is. Presumably, since this contribution is constant irrespective of the aerosol loading, depending upon the background signal amplitude, this contribution could be more important at lower signal levels. Are we talking about Sbgcorr = 0.1 Mm^-1 or 1 Mm^-1 or 10s Mm^-1. I would like to have my fears allayed that in the limit of weak aerosol absorption signal (the atmospherically relevant situation) that the reported PAS signal for the aerosol is not the result of subtracting two big numbers to get a small number.

Response:

    The reviewer raises a very important point. The background correction is indeed constant and is invariant with aerosol/ozone concentration. $S_{corr}^{bg}$ is a corrected raw microphone response and has arbitrary values (not units of Mm$^{-1}$). In atmospheric measurements (from an aircraft) of aerosol, the sensitivity of absorption measurements depends on absorption strength and this background response introduces uncertainties in the measured aerosol absorption of 0.2, 2.0 and 20.5% at 100, 10 and 1 Mm$^{-1}$ absorption strengths, respectively (see Davies *et al.*, Atmos. Meas. Tech. Discuss., 2019). Importantly for this work, $S_{corr}^{bg}$ is typically <10% of the raw photoacoustic signal during ozone calibrations. We have added a statements on line 9 of page 9 to state:

*$S_{corr}^{bg}$ is constant over a calibration and typically represents <10% of the photoacoustic signal during calibrations with ozone.*

At the risk of being a bit pedantic, could you please reorder the legend on Figure 3b. As currently displayed, the traces are solid, dotted, and dashed, while the legend is solid, dashed, and dotted. This is certainly a cosmetic request, but makes it easily for the reader. Also, are error bars present on the PAS signal (and are just too small to be observed on the plot scale) or are they not present. If the latter, please add.

Response:

We thank the review for suggesting reordering the legend, which we have now done. With regards to vertical error bars, these error bars are already present on the PAS sensitivity plots (Figure 3). However, we had overlooked a statement to describe the error bars included in the figure legend. Therefore, in the figure legend for Figure 3, we have included the following statement:

*The measured data include vertical error bars that represent one standard deviation in the measured sensitivity, although these error bars are not visible on the vertical scale shown. Horizontal error bars represent the uncertainty in $O_2$ mass fraction arising from the standard errors in the mass flow controller flow rates that control concentrations of $O_2$ and $N_2$ in the bath gas.*

This reviewer is most intrigued by the potential consequences of reduced pressure at aircraft sampling altitudes on quenching rates. The authors kick this can down the road but this could be quite interesting from both a fundamental energy transfer perspective as well as a practical aspect. Staying with this theme, I cannot help but wonder if NO2 as a calibration gas - available with either N2 or air as the bath gas - might also exhibit a similar quenching sensitivities to differing bath gas mixtures as displayed by ozone. As highlighted above, one would expect that bath gas quenching could shift the quantum yield for NO2 photodissociation at 405 nm.

Response:

We agree with the reviewer that the pressure dependence in ozone quenching rates is of great interest. Indeed, we state on page 19 line 19 that studying the pressure dependence in this quenching is the subject of ongoing work. With regards to $NO_2$, we refer the reviewer to work by Kalkman and Kesteren (DOI: 10.1007/s00340-007-2895-0) who show that the quenching is maximised for a bath gas in the limit of pure $O_2$, although it is difficult to ascertain whether this maximum corresponds to complete quenching of energy into translational degrees of freedom for generation of a photoacoustic signal.

The last request, which the authors are encouraged to do, but certainly not required given the target audience. It would be nice to add a figure displaying the potential energy surfaces (even as a simple 2-D plot) for the various O3 dissociation pathways. This is certainly the chemical physicist in me requesting this as that is how I can readily see what is going on. For those researchers not accustomed to chemical physics, such a figure would help clarify the subtle pathways present in ozone.

Response:

This was certainly tempting when we were writing the manuscript; indeed, three of the co-authors have chemical physics backgrounds. Instead, we refer the reader (Page 11, lines 23 – 27) to the work of Grebenschikv *et al.* (2007) to inspect the potential energy surfaces along the $O_2$-O dissociation coordinate. We prefer to point the reader to the relevant literature and not repeat these readily-available figures in this contribution, particularly given the target audience.

[revised manuscript text omitted]